# Risk for short-term undesirable outcomes in older emergency department users: Results of the ER² observational cohort study

**Cyrille P. Launay**[1,2]*, **Kevin Galery**[1,2], **Christine Vilcocq**[1,2], **Marc Afilalo**[3], **Olivier Beauchet**[1,4,5,6]

**1** Department of Medicine, Division of Geriatric Medicine, Sir Mortimer B. Davis—Jewish General Hospital and Lady Davis Institute for Medical Research, McGill University, Montreal, Quebec, Canada, **2** Centre of Excellence on Longevity of McGill Integrated University Health and social services Network, Quebec, Canada, **3** Emergency Department, Jewish General Hospital, McGill University, Montreal, Quebec, Canada, **4** Departments of Medicine, University of Montreal, Montreal, Quebec, Canada, **5** Research Center of the Geriatric University institute of Montreal, Montreal, Quebec, Canada, **6** Lee Kong Chian School of Medicine, Nanyang Technological University, Singapore, Singapore

* cyrille.launay@mcgill.ca

**Data Availability Statement:** Data and materials are available on request. This study is a secondary use of an already constituted database. If a researcher wants to access to data, according to

## Abstract

### Background

The "Emergency Room Evaluation and Recommendations" (ER²) is a clinical tool designed to determine prognosis for the short-term Emergency Department (ED) undesirable outcomes including long length of stay (LOS) in ED and in hospital, as well as the likelihood of hospital admission during an index ED visit. It is also designed to guide appropriate and timely tailor-made geriatric interventions. This study aimed to examine whether ER² assessment part was: 1) usable by ED healthcare workers (*e.g.* nurses) and 2) scoring system associated with long LOS in ED and in hospital, as well as hospital admission in older ED users on stretchers.

### Methods

Based on an observational, prospective and longitudinal cohort study 1,800 participants visiting the ED of the Jewish General Hospital (Montreal, Quebec, Canada) were recruited between September and December 2017. ER² assessment determined three risk-levels (*i.e.*, low, medium and high) for short-term ED undesirable outcomes. The rate of ER² digital form completed, the time to fill ER² items and obtain ER² risk-levels, the LOS in ED and in hospital, and hospital admission were used as outcomes.

### Results

ER² was usable by ED nurses in charge of older ED users. High-risk group was associated with both increased ED stay (coefficient of regression β = 3.81 with P≤0.001) and hospital stay (coefficient of regression β = 4.60 with P = 0.002) as well as with hospital admission (HR = 1.32 with P≤0.001) when low ER² risk level was used as referent level. Kaplan-Meier distributions showed that the three risk groups of participants differed significantly (P =

the protocol of the databank, he needs to write a full protocol of secondary use of a databank, obtain an approval from their own ethic committee and contact the manager of the databank (corresponding author): Olivier Beauchet, MD, PhD; Department of Medicine, Division of Geriatric Medicine, Sir Mortimer B. Davis - Jewish General Hospital, McGill University, 3755 chemin de la Côte-Sainte-Catherine, Montréal, QC H3T 1E2, Canada; E-mail: olivier.beauchet@mcgill.ca As Olivier Beauchet is the manager of the databank, all requests for data access have to be sent to him. Request should be sent to: Julia Chabot, MD ; Department of Medicine, Division of Geriatric Medicine, St. Mary's Hospital Center, 3830 Lacombe Avenue, Montreal, Quebec H3T 1M5; Email: julia.chabot@mcgill.ca. All requests needs a cover letter explaining the objective, justification and the referent ethic committee.

**Funding:** The study was self funded by the Centre of Excellence on Longevity, through funds allocated to research by the Foundation of the Jewish General Hospital. The authors received no specific funding for this work. The funders had no role in study design, data collection and analysis, decision to publish, or preparation of the manuscript.

**Competing interests:** No, the authors have declared that no competing interests exist.

0.001). Those with high-risk level ($P \leq 0.001$) were discharged later from hospital to a non-hospital location compared to those with low risk. There was no significant difference between those classified in low-risk and in medium-risk groups (P = 0.985) and those in medium and high-risk groups (P = 0.096).

## Conclusion

The ER$^2$ assessment part is usable in daily practice of ED care and its risk stratifications may be used to predict adverse outcomes including prolonged LOS in ED and in hospital as well as hospital admission.

## Trial registration

NCT03964311

## 1. Introduction

Older (i.e., age$\geq$75) patients visiting Emergency Department (ED) account up to 25% of all ED users [1, 2]. Chronic diseases and geriatric syndromes strongly impact older ED users' abilities and expose them to a greater risk for short-term undesirable outcomes including long length of stay (LOS) in ED and in hospital or hospital admission, compared to younger ED users [3, 4]. These undesirable outcomes largely account for EDs overcrowding and are therefore increasingly challenges facing hospitals [5].

One way for preventing or reducing the short-term undesirable outcomes is 1) to identify ED users with a high risk for their occurrence using a screening tool and 2) to provide timely and tailor-made geriatric interventions [2, 6, 7]. Comprehensive geriatric interventions in EDs have been associated with better clinical outcomes such as objective improvement in functional status [7]. Most of the clinical tools described in the literature that are designed for this purpose assess the risk for undesirable outcomes occurring after the discharge from EDs or from hospital, except one tool known as "Emergency room evaluation and recommendations" (ER$^2$), which provides levels of risk for short-term undesirable outcomes during an ED index visit [2, 8, 9]. ER$^2$ has been developed in France and is comprised of two parts: an assessment part and an interventional part. The ER$^2$ assessment part provides a risk stratification in three levels: low, medium and high [10–15]. Recently, a study performed in Canada in older patients admitted to a geriatric ward after an ED visit demonstrated that patients with ER$^2$ high and medium risk levels had significant longer hospital stays compared to those with a low-risk level [16].

In addition to good predictive performance, usability and effectiveness are major characteristics of a screening tool dedicated to EDs. Delay and over capacity are chronic conditions in EDs [3–5]. EDs are often cited as stressful environments, with increasing volume and acuity of ED presentations resulting in high pressure and high-volume workloads that may compromise their practice and quality of care [1–5]. Therefore, a change in practice of ED healthcare workers by implementing a new clinical tool such as ER2 needs to take into consideration the usability of the proposed tool, in order to adapt to an institution's specific conditions. Usability is a measure of how well a specific user in a specific context can use a product, in our case ER2, to achieve a defined goal efficiently and satisfactorily [17]. A tool's usability depends on how well its features accommodate users' needs and contexts. A clinical tool usable in daily practice

of ED requires ease of use, efficiency (such that users can quickly complete the tool in a timely manner) and effectiveness (i.e., it supports users in completing actions accurately) [18].

Usability and prognostic value of $ER^2$ assessment part for short-term undesirable outcomes in older ED Canadian users remains to be determined. We hypothesized that $ER^2$ assessment could be used by ED staff and its risk stratification would be associated with LOS in ED and hospital as well as hospital admissions. A systematic review on feasibility of frailty screening tools in EDs highlighted that completion rate and time elapsed were the most appropriate criteria to assess usability [18]. In order to determine usability in our centre more precisely, we proposed to further define time elapsed as the time to complete the $ER^2$ assessment form and to record its items in the patient's digital file, as well as to assess the evolution of both the $ER^2$ completion rate and the average time elapsed over the first months of its implementation in the ED daily practice. It is postulated that an observed decrease in time to complete the $ER^2$ assessment and record its items in the patient's digital file is a surrogate measure of ease to learn and usability, and that a high completion rate and an observed increase with time are surrogate makers of efficiency and effectiveness. This study aims to examine whether the $ER^2$ assessment part was: 1) usable by ED healthcare workers (e.g. nurses) and 2) scoring system associated with long LOS in ED and hospital, as well as hospital admission in older ED users on stretchers.

## 2. Materials and methods

### 2.1 Study design and population

This observational, prospective and longitudinal cohort study was conducted in the ED of the Jewish General Hospital (Montreal, Quebec, Canada) between September 1st to December 31st 2017. The criteria of inclusion were age $\geq$ 75, an unplanned ED visit, being on a stretcher and agreement to participate in the study. The exclusion criteria were a concomitant participation in an experimental study and the occurrence of death during the hospitalization. During the 4 month-recruitment period, 5,605 older ED users visited the ED. Among this group, 4,724 (84.3%) were on a stretcher and 1,800 (38.1%) were assessed with $ER^2$. There were significant differences between older ED users with and without an $ER^2$ assessment. Compared to older ED users without an $ER^2$ assessment, those with an $ER^2$ assessment were older (P≤0.001), more frequently institutionalized (*e.g.* living in nursing homes) (P≤0.001), more frequently presenting with organ failure and had less neuropsychiatric disorders (P≤0.001), stayed for a longer duration in ED (P≤0.001) and were more frequently admitted to hospital (P≤0.001) (please see complementary data).

### 2.2 Emergency room evaluation and recommendations

The $ER^2$ assessment is composed of six close-ended format questions (*i.e.*, yes *versus* no): Aged ($\geq$ 85), male, polypharmacy ($\geq$ 5 different medications per day), use of formal (health care or social services) and/or informal (family and/or friend) home support, use of a walking aid (regardless its type), and temporal disorientation (defined as inability to correctly identify the current month and/or year). A score of five points is assigned to items "use of walking aid" and "temporal disorientation" (major criteria), whereas for the other items the assigned score is one point (minor criteria). The weighting of points for $ER^2$ items is based on results of previous studies [10–16, 19]. The scoring range is from 0 (lowest risk) to 14 (highest risk). The $ER^2$ assessment score stratifies risk for short-term undesirable outcomes in three levels: low, medium and high. The low-risk group is defined by the combination of 3 minor criteria or less among age $\geq$ 85, male, polypharmacy, and use of home support. The score in this group ranged from 0 to 3. The medium risk is characterized by one major criteria (i.e., temporal

disorientation or use of walking aid) or the combination of the four minor criteria (i.e., age $\geq$ 85 + male gender + polypharmacy + use of home support) and is defined by a score ranging from 4 to 5. Finally, the high-risk group is defined by a score $\geq$ 6. This implies that either both major criteria are met (i.e., the patient has both temporal disorientation and necessitates a walking aid), or the presence of one major criterion and at least one minor criteria. All ED healthcare workers (*i.e.*, nurses, physicians, social workers, physiotherapists, coordinators) were blinded of the ER$^2$ assessment score and risk stratification.

## 2.3 Baseline assessment of participants' characteristics

Upon their arrival to the ED, participants had an assessment performed by the nurse in charge of triage. This baseline assessment collected information regarding: age, sex, and place of living prior to ED visit categorized in three types including home, nursing home (*i.e.*, a facility for the residential care of elderly or disabled people), and transfer from another hospital or other healthcare institution such as a rehabilitation centre (when the patient is transferred in the context of acute disease). In addition, hospital health administrative areas, which is specific to Quebec's healthcare system and refers to the hospital they are assigned to according to where they live, has also been recorded. The Canadian ED Triage and Acuity Scale was performed to avoid a confusion bias on the level of severity of the patients [20]. Reasons for ED visits and associated level of severity are particularly variable among older patients and may strongly impact the LOS in EDs and hospital, as well as rate of hospital admissions [2]. Triage is a process during which patients are prioritized and classified according to the type and urgency of their health condition. Triage is the first step of ED visit assesses the type and severity of patient health conditions, determines access to appropriate treatments and assigns appropriate human health resources. This scale is composed of 5 levels of urgencies which are: level 1, defined as resuscitation; level 2, defined as emergent; level 3, defined as urgent; level 4, defined as less urgent; and level 5, defined as non-urgent. ED physicians recorded the primary reason for ED visit in patients' digital file. This information was extracted from patients' digital file database and categorized in 5 sub-types: Organ failure, defined as an acute organ decompensation; mobility disorders, defined as gait and/or balance impairment and/or fall with or without fall-related injuries; Neuropsychiatric disorders, defined as delirium, dementia, behavioral disorders; cancer, defined as a group of diseases involving abnormal cell growth with the potential to invade or spread to other parts of the body; social issue, defined as the absence of symptoms of acute disease combined with an acute increase in the use of formal and/or informal home and social services leading to an inability to cope; and miscellaneous reasons not included in the previous categories. Once triage is complete and the ED user is on a stretcher, the assigned nurse performed ER$^2$ assessment at bedside.

## 2.4 Follow-up

The information regarding the number of ER$^2$ assessments completed, time elapsed to complete the assessment, LOS in ED and in hospital, admissions to hospital and date of discharge were extracted from the patients' digital file database. LOS was calculated from the administrative registry and corresponded to the delay in hours in ED (*i.e.*, time of ED arrival to time to ED discharge), and in days for hospital stay (*i.e.*, the day of ED arrival to the day of hospital discharge).

## 2.5 Outcomes

The outcomes examining usability of ER$^2$ assessment were: 1) The rate of ER$^2$ completed using the ratio: Amount of older ED users with a ER$^2$ score / Total amount of older adults who

visited ED; 2) The time to fill and record $ER^2$ items in the patient's digital file calculated automatically with the patient's digital time code and corresponding to the delay (in minutes) between the arrival at ED and the validation of $ER^2$ items in the $ER^2$ software. Rate of $ER^2$ assessment completed as well as time to fill and record $ER^2$ were compared month by month. Naturally, efficient integration of a screening tool in clinical routine requires a training period. The outcomes examining the short-term undesirable outcomes were: 1) LOS in the ED, expressed in hours; 2) LOS in hospital, expressed in days; 3) The hospital admission rate using the ratio: amount of older ED users admitted to hospital /total amount of older adults who visited ED.

## 2.6 Standard protocol approvals, registrations, and participant consents

The $ER^2$ study was classified as a clinical quality improvement program for older ED users care plan by the Ethic Committee and the West-Central Montreal Health Review Office for quality programs of the Jewish General Hospital (Montreal, Quebec, Canada). Verbal informed consent was obtained for all participants following a systematic and standardized process used in the ED ward where the study was performed. Participants, or their legal guardian when appropriate, were informed that their medical information may be used for research purposes. If they disagreed, they informed the physician taking care of them and a note was recorded in their chart. The Ethics Committee the Jewish General Hospital approved this process.

## 2.7 Statistical analysis

The participants' baseline characteristics were summarized using means, standard deviations (SD), frequencies and percentages as appropriate. First, comparisons of rate of $ER^2$ assessment completed each month and time to complete it were performed using Chi-square or unpaired t-test. Second, participants with a $ER^2$ score were separated in three groups based on $ER^2$ three risk-levels (*i.e.*; low, medium, high) and between group-comparisons of participant's characteristics were performed using an analysis of variance (ANOVA) with Bonferroni correction for multiple comparisons. Third, regression models were performed to examine the association of the LOS in ED and in hospital (linear regression) as well as hospital admission (Cox regression) used as dependent variables with separated model for each outcome with Emergency Room Evaluation and Recommendation risk-levels used as independent variables. The low-risk level was used as the reference group. All regressions models were adjusted for hospital health areas (*i.e.*, patients in Quebec are assigned to a hospital according to where they live), localization before ED visit, reasons for ED visit and Canadian emergency department triage and acuity scale level. Fourth, the elapsed time to discharge from hospital to a non-hospital location by survival Kaplan-Meier curves and log-rank test were also performed. Participants were not included if they died during their hospitalization. P-values $<0.05$ were considered statistically significant. All statistics were performed using SPSS (version 23.0; SPSS, Inc., Chicago, IL).

## 2.8 Patient and public involvement

Patients and the public had no input into decisions regarding the research question, outcome measures, study design, recruitment and conduct of the study. Information regarding the burden of the intervention or time required to participate in the research was not provided to patients and the public. The patients and the general public were not involved in the conception or publication of this research project.

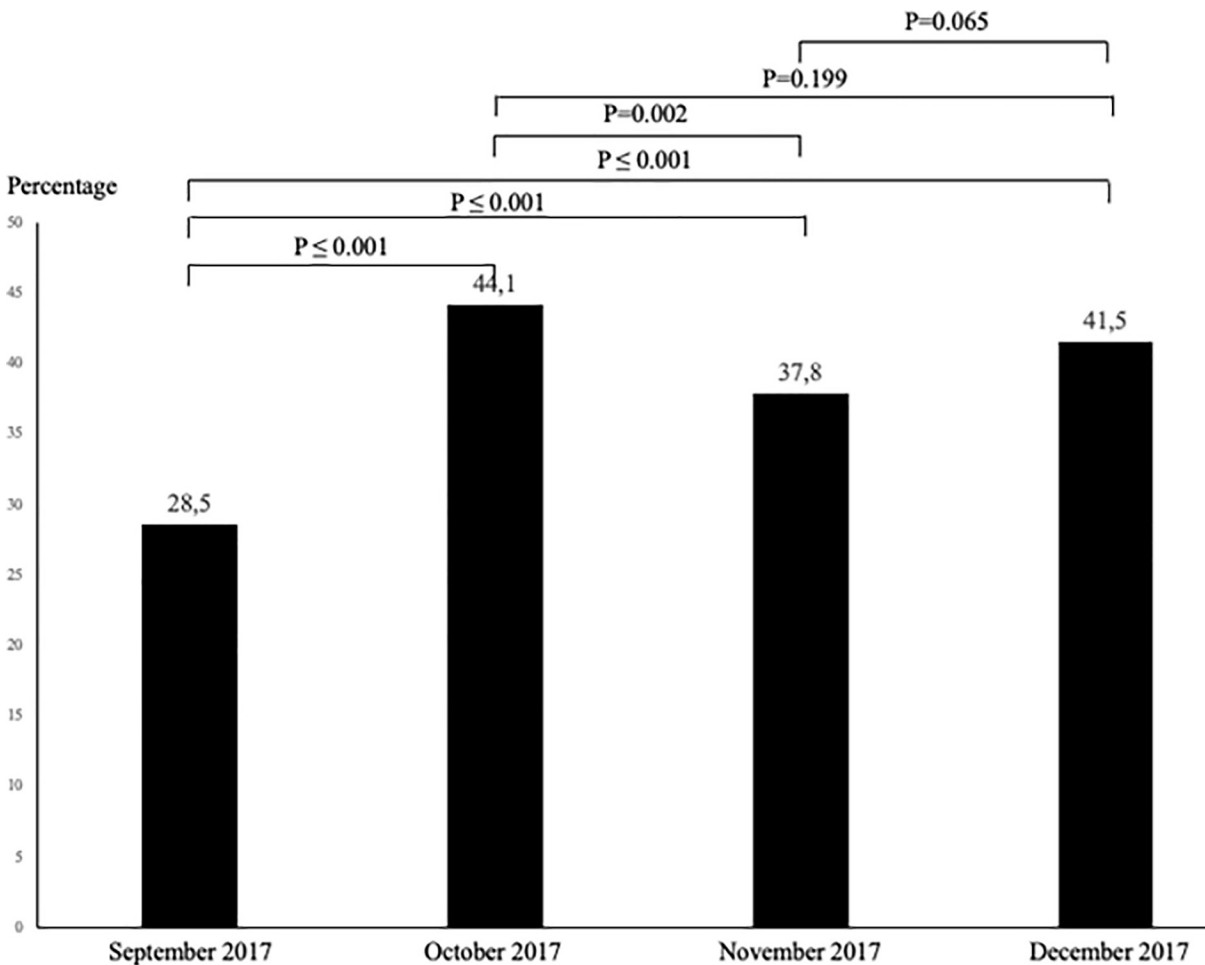

**Fig 1. Evolution of the percentages of ER$^2$ assessment completed (*i.e.*, amount of older ED users with a ER$^2$ score / total amount of older adults who visited ED) from September to December 2017 (n = 1,800).**

## 3. Results

The Fig 1 shows the evolution of the percentages of ER$^2$ assessment completed from September to December 2017. There was an increase of this percentage with significant difference between September and all other months (P≤0.001). There was no other significant difference between months, except between October and November 2017 where the percent decrease from October to November (P = 0.002).

Time to fill and record ER$^2$ items in the patient's digital file was higher in September compared to the other months (P≤0.034) and there was no significant difference between months for all other comparisons (Fig 2).

Table 1 reports the comparison of participants' baseline characteristics between participants separated in three groups based on their risk-levels (*i.e.*, low, medium and high). Those with a low-risk level were younger compared to those with medium and high-risk levels (P≤0.001). There were more females in the low-risk group compared to the medium -risk group (P≤0.001), but less compared to high-risk group (P = 0.038). There were more females in the high-risk group compared to the medium -risk group (P≤0.001). Participants in the low-risk group were less frequently transferred from other hospital (P≤0.001), more frequently from

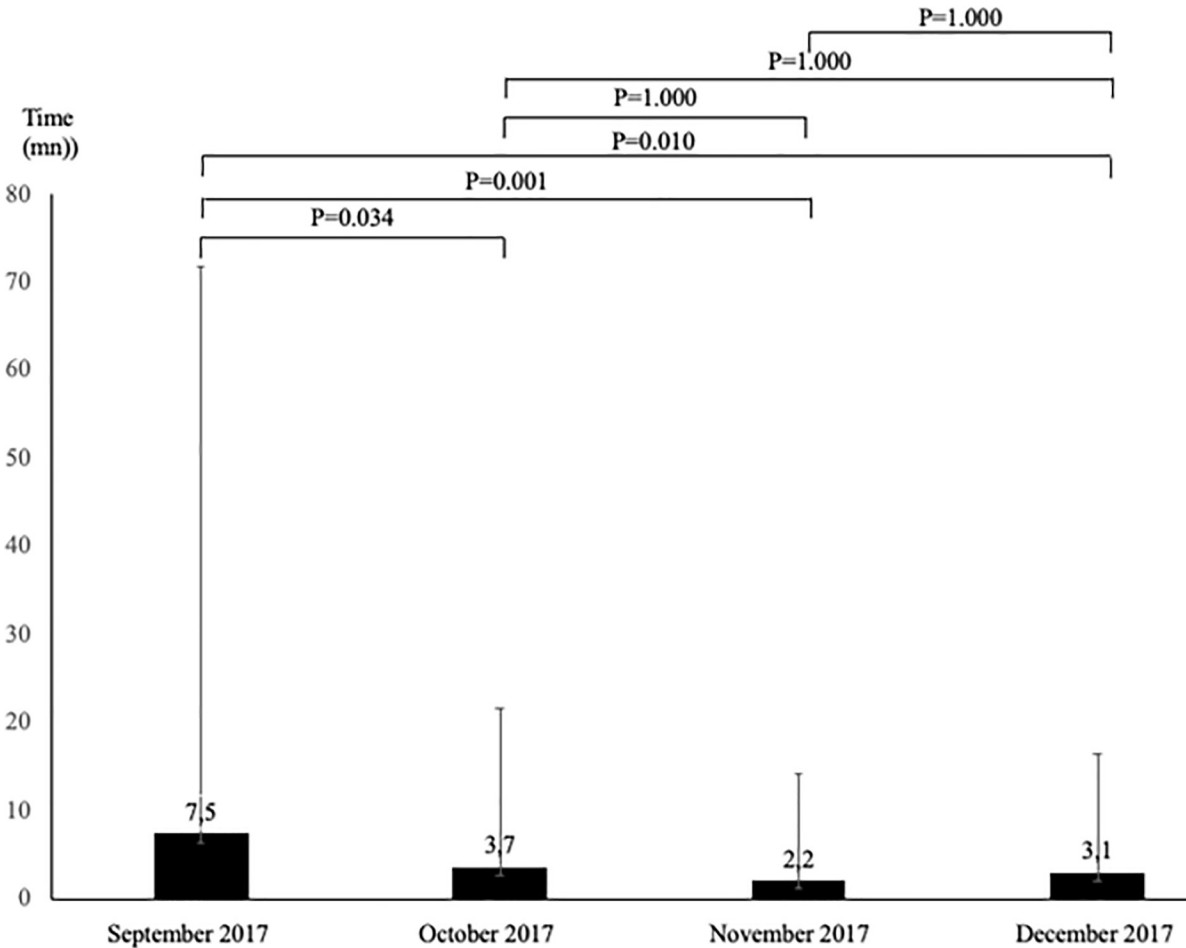

**Fig 2. Evolution of time to fill ER$^2$ assessment from September to December 2017 (n = 1,800).**

home and less from nursing homes (P≤0.001) or rehabilitation centre (P = 0.023) compared to those with high-risk level. Participants in the high-risk group came less frequently from home and more from nursing homes (P≤0.001) compared to those with medium-risk level. Neuropsychiatric disorders as the primary reason for ED visit were less frequent in medium-risk group compared to the other groups (P≤ 0.022). There were more patients with a Canadian ED Triage and Acuity Scale level 1 in the high-risk group compared to the low-risk group (P = 0.016), as opposed to the medium-risk group, which has a significantly greater number of patients stratified as level 2 (P = 0.022) and level 4 (P = 0.019) compared to those in high-risk group. In addition, there were more patients in medium-risk group with level 2 compared to those in the low-risk (P = 0.020) and high-risk group (P = 0.001).

The length of stay in ED and in hospital, as well as the frequency of hospital admissions were higher in the high-risk group (Table 2) compared to patients in the low-risk group (P≤0.001).

In addition, the length of stay in ED (coefficient of regression β = 3.81 with P≤0.001) and in hospital (coefficient of regression β = 4.60 with P = 0.002) were longer in the high-risk group compared to the low-risk group.

When low ER$^2$ risk level was used as referent level, high-risk level was associated with both increased ED stay (coefficient of regression β = 3.81 with P≤0.001) and hospital stay (coefficient of regression β = 4.60 with P = 0.002) as well as with hospital admission (HR = 1.32 with

**Table 1. Comparison of baseline characteristics of participants on stretcher visiting emergency department who had Emergency Room Evaluation and Recommendation (ER²) assessment completed separated according to their risk-level for short-term undesirable outcomes (n = 1,800).**

| | ER² risk-level | | | P-Value* | | | |
|---|---|---|---|---|---|---|---|
| | Low | Medium | High | Overall | Low *versus* | | Medium |
| | (n = 637) | (n = 64) | (n = 1099) | | Moderate | High | *versus* high |
| Age (years), mean±SD | 83.0±5.3 | 86.4±5.2 | 87.1±6.2 | ≤**0.001** | ≤**0.001** | ≤**0.001** | 1.000 |
| Female, n (%) | 356 (55.9) | 13 (20.3) | 670 (61.0) | ≤**0.001** | ≤**0.001** | **0.038** | ≤**0.001** |
| Localization before ED visit, n (%) | | | | | | | |
| Hospital Health areas† | 322 (50.5) | 33 (51.6) | 668 (60.8) | ≤**0.001** | 0.887 | ≤**0.001** | 0.143 |
| Place of living | | | | | | | |
| Home | 473 (74.3) | 48 (75.0) | 581 (52.9) | ≤**0.001** | 0.896 | ≤**0.001** | **0.001** |
| Nursing home | 63 (9.9) | 10 (15.6) | 383 (34.8) | ≤**0.001** | 0.152 | ≤**0.001** | **0.002** |
| Hospital‡ | 98 (15.4) | 6 (9.4) | 135 (12.3) | 0.120 | 0.197 | 0.068 | 0.488 |
| Other‖ | 3 (0.5) | - | - | 0.064 | 0.582 | **0.023** | - |
| Reason for ED visit, n (%) | | | | | | | |
| Organ failure¶ | 361 (56.7) | 42 (65.6) | 591 (53.8) | 0.118 | 0.167 | 0.243 | 0.064 |
| Mobility disorders§ | 91 (14.3) | 13 (20.3) | 182 (16.6) | 0.282 | 0.196 | 0.210 | 0.435 |
| Neuropsychiatric disorders# | 66 (10.4) | 1 (1.6) | 140 (12.7) | **0.013** | **0.022** | 0.140 | **0.008** |
| Cancer | 19 (3.0) | 2 (3.1) | 39 (3.5) | 0.815 | 0.949 | 0.527 | 0.858 |
| Social issue** | 85 (13.3) | 5 (7.8) | 130 (11.8) | 0.356 | 0.207 | 0.356 | 0.330 |
| Other†† | 15 (2.4) | 1 (1.6) | 17 (1.5) | 0.475 | 0.686 | 0.228 | 0.992 |
| Canadian ED Triage and acuity scale, n (%) | | | | | | | |
| Level 1 –Resuscitation | 8 (1.3) | 1 (1.6) | 34 (3.1) | **0.049** | 0.835 | **0.016** | 0.486 |
| Level 2 –Emergent | 180 (28.3) | 27 (42.2) | 256 (23.3) | **0.001** | **0.020** | **0.022** | **0.001** |
| Level 3 –Urgent | 338 (53.1) | 27 (42.2) | 574 (52.2) | 0.252 | 0.097 | 0.738 | 0.118 |
| Level 4 –Less urgent | 104 (16.3) | 9 (14.1) | 230 (20.9) | **0.037** | 0.639 | **0.019** | 0.186 |
| Level 5 –Non-urgent | 7 (1.1) | - | 5 (0.5) | 0.226 | 0.399 | 0.119 | 0.589 |
| Length of stay, mean±SD | | | | | | | |
| Emergency department (hours) | 19.8±13.2 | 23.2±16.1 | 23.8±13.4 | ≤**0.001** | 0.181 | ≤**0.001** | 1.000 |
| Hospital (days) | 11.6±14.7 | 12.1±19.7 | 15.7±20.7 | ≤**0.001** | 1.000 | **0.009** | 0.822 |
| Hospital admission, n (%) | 269 (42.2) | 34 (53.1) | 609 (55.4) | **0.009** | 0.093 | ≤**0.001** | 0.720 |

ER²: Emergency Room Evaluation and Recommendation; SD: Standard deviation; ED: Emergency Department

†: Patient living in the referent hospital health areas (Hospital health areas" is specific to Quebec's healthcare system and refers to the hospital they are assigned to, according to where they live)

*: Comparisons based on analysis of variance with Bonferroni correction or chi square, as appropriate

‡: Patients transferred from another Emergency Department hospital

‖: Patients transferred from rehabilitation centre

¶: Defined as an acute organ decompensation

§: Defined as gait and/or balance impairment and/or fall with or without fall-related injuries

#: Defined as delirium, dementia, behavioral disorders

**: Defined as the absence of symptoms of acute disease combined with an acute increase of the use of formal and/or informal home and social services leading to an inability to cope at home

††: all other reasons not included in the previous categories; P value significant < 0.0023 because of multiple comparisons (n = 21) indicated in bold.

P≤0.001). Finally, Kaplan-Meier distributions showed that the three risk groups of participants differed significantly (P = 0.001). Those with high-risk level (P≤0.001) were discharged later from hospital to a non-hospital location compared to those with low-risk (Fig 3). There was no significant difference between those classified in low-risk and in medium-risk groups (P = 0.985) and those in medium and high-risk groups (P = 0.096).

**Table 2. Regression models showing the association of the length of stay in emergency department and hospital (linear regression) as well as hospital admission (Cox regression) used as dependent variables with separated model for each outcome with Emergency Room Evaluation and Recommendation risk-levels used as independent adjusted by participants' baseline characteristics in older Emergency Department users on stretcher (n = 1,800).**

| | Linear Regression | | | | | | Cox regression | | |
| --- | --- | --- | --- | --- | --- | --- | --- | --- | --- |
| | Length of stay | | | | | | | | |
| | Emergency department | | | Hospital | | | Hospital admission | | |
| | β | [95%CI] | P-Value | β | [95%CI] | P-Value | HR | [95%CI] | P-Value |
| Model ER$^2$ | | | | | | | | | |
| Risk level*† | | | | | | | | | |
| Low | Ref. | | | Ref. | | | Ref. | | |
| Medium | 1.78 | [-1.73;5.29] | 0.319 | 2.82 | [-4.18;9.81] | 0.790 | 1.17 | [0.82;1.69] | 0.389 |
| High | 3.81 | [2.42;5.20] | **≤0.001** | 4.60 | [1.70;7.49] | **0.002** | 1.32 | [1.13;1.54] | **≤0.001** |

ER$^2$: Emergency Room Evaluation and Recommendation; β: Coefficient of regression beta; Ref: the low-risk level was used the reference group

*: Model adjusted on age, sex, localization before Emergency Department visit, reason for Emergency Department visit and Canadian Emergency Department Triage and acuity scale level

†: Low-risk level used the referent level

## 4. Discussion

The findings show that ER$^2$ assessment part was usable by ED nurses caring for older ED users on stretcher and that ER$^2$ risk stratification was significantly associated with LOS and hospital admissions. Low-risk level was associated with a short LOS and a low rate of hospital admissions, whereas high-risk level was associated with long LOS and a high rate of admission to hospital.

Our study demonstrates usability of ER$^2$ by ED nurses. Even if the rate of completion of ER$^2$ was approximately 40%, a significant increase of this rate was observed during the 4-month period of the study. This finding suggests that there is a learning curve and that ED nurses may require a training period to implement ER$^2$ in their daily practice. A recent systematic review showed that most tools have not considered usability in their development, with the exception of the Identification of Senior At Risk (ISAR) [18]. It showed that 37% of older ED users were assessed using ISAR, which is a similar completion rate to ER$^2$ in the present study [21]. It is important to note that ER$^2$ study was classified as a clinical quality improvement program for older ED users. This choice was made to consider a *"real life condition"* of ED practice, in order to ensure that ER$^2$ could be feasibly integrated in the daily practice of ED nurse staff. Over the course of the study, the time to complete the ER$^2$ decreased to achieve an average of 3 minutes. In order for a screening tool to be considered usable in the ED setting, it must be completed in less than 5 minutes [20]. ER$^2$ has been designed for daily practice in ED and its items depend on both clinical and objective information that are easily measurable during a short physical examination and do not depend on the patient or family's answers [2, 8–15, 19].

ER$^2$ risk levels were associated with LOS in ED and in hospital, low-risk being associated with short LOS and high-risk with long LOS. This result is consistent with previous publications [8–15, 19]. This information may help ED staff manage their older patients more efficiently. It suggests that older ED users who will stay a shorter period of time in ED or in hospital have a phenotype of middle-old patients (i.e., less likely to be oldest old patients) with lower frailty level and related less severe acute diseases [22]. Compared to middle-old patients, oldest old patients are known to be at higher risk for prolonged length of stay in ED and high rate of admission to hospital [23]. The ER$^2$ tool allows for more effective triage of older ED

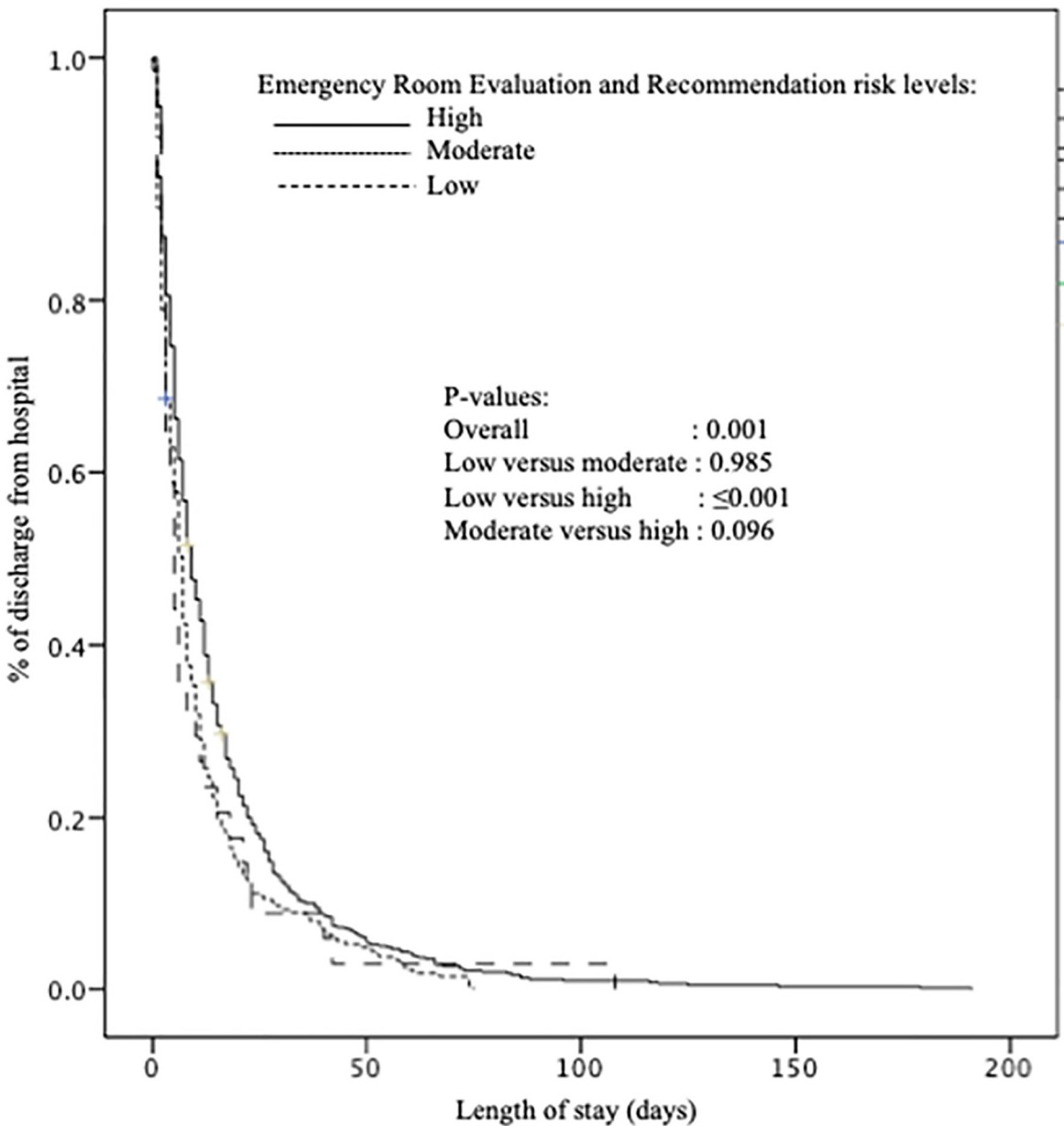

**Fig 3. Kaplan-Meier estimates of the probability of discharge from hospital to a non-hospital location based on the ER² risk-levels (i.e.; low, medium, high) (n = 1,800).**

patients; it enables ED healthcare workers to identify this lower risk population amongst elderly ED patients and to direct the time and resources to the elderly patients who require more care [2]. The main characteristic of older ED users compared to younger users is an accumulation of severe and chronic morbidities and related-disabilities [2, 3]. These particular health and functional conditions greatly influence the ED care plan [2–4]. Although older adults undergo more diagnostic tests and procedures than younger ED patients, their diagnoses tend to be less accurate [2, 3]. This has been attributed to the complex interplay between acute and chronic diseases that account for atypical clinical presentations, which poses an immense challenge in the busy ED setting with regards to providing accurate diagnoses and

choosing appropriate disposition and care plans [2–4, 8–12]. Substantial resources are devoted to promptly identify and treat acute diseases in EDs, but multi-morbidities and disabilities are more likely to be ignored and undertreated [1–5]. Thus, they are more prone to poor short-term outcomes defined as long ED and hospital LOS, high hospital admission rate and significantly higher rates of in-hospital mortality [2–4, 8–15, 19].

The present study has limitations to consider. Firstly, the study involved a single centre which limits the external validity of the results. The studied population in our study may not be representative of all older ED users. Secondly, although adjustment for clinical characteristics influencing the studied short-term undesirable outcomes was performed, there are still potential residual confounders which may influence the LOS. Thirdly, $ER^2$ does not take into consideration the reasons of ED visits and its severity which is usually an acute disease directly influencing the occurrence of short-term ED adverse events. For instance, an acute disease may decompensate in a cascade of complications related to chronic morbidities and geriatric syndromes accumulated in older ED users, leading to multiple acute organ decompensations and disabilities, which predisposes older patients to long length of stays and hospitalisations. However, in all of the regression models used to examine the association of between $ER^2$ risk levels and the short-terms ED undesirable events, there was an adjustment for reasons of ED visits and its severity using the Canadian ED Triage and acuity scale was performed [20].

In conclusion, $ER^2$ assessment is usable in daily practice of ED care by nurses and risk-level stratifications were significantly associated with LOS and hospital admission, low-risk being associated with shorter LOS and hospital admission and high-risk being associated with longer LOS and a high rate of admission. There is a need to confirm these results with multicentre observational cohort studies in Canadian hospitals.

## Acknowledgments

The authors are grateful for their cooperation: 1) the participants; 2) nurses of the Department of emergency of Jewish General hospital, and in particular Mrs Valerie Shneidman and Mr Jonathan Harroche; 3) Pharmacists of Jewish General hospital, and in particular Mrs Julie Roy and Mrs Nikki Kampouris; 4) Mrs. Betty Elkaim (Vice-President and Chief Development Officer of the Foundation of the Jewish General Hospital); 5) the Department of information and technology of Jewish General Hospital, and in particular Mrs Isabelle Aumont, Mrs Maria Veres, Mrs Christine Bougie and Mrs France Guimont; 6) Mr Joshua Lubov that provided a precious revision of the manuscript and 7) Mrs. Abbigail Shaw and Mr. Davis Schiller.

## Author Contributions

**Conceptualization:** Cyrille P. Launay, Marc Afilalo, Olivier Beauchet.

**Formal analysis:** Olivier Beauchet.

**Investigation:** Cyrille P. Launay.

**Methodology:** Cyrille P. Launay, Olivier Beauchet.

**Supervision:** Cyrille P. Launay, Kevin Galery, Christine Vilcocq, Olivier Beauchet.

**Writing – original draft:** Cyrille P. Launay, Marc Afilalo, Olivier Beauchet.

**Writing – review & editing:** Cyrille P. Launay, Kevin Galery, Christine Vilcocq, Marc Afilalo, Olivier Beauchet.

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
