## [Decision Letter · Decision Letter 0]

3 Sep 2020

PONE-D-20-15983

Risk for short-term adverse events in older emergency department users: Results of the ER2 observational cohort study

PLOS ONE

Dear Dr. Launay,

Thank you for submitting your manuscript to PLOS ONE. After careful consideration, we feel that it has merit but does not fully meet PLOS ONE’s publication criteria as it currently stands. Therefore, we invite you to submit a revised version of the manuscript that addresses the points raised during the review process.

The editor has also reviewed the manuscript and agree wit reviewers. The authors are strongly encouraged to read and consider the comments. If there is a decision to resubmit, the authors need to respond thoughtfully nd specifically to make consideration for publication possible.

We look forward to receiving your revised manuscript.

Kind regards,

Joseph Telfair, DrPH, MSW, MPH

Academic Editor

PLOS ONE

Journal Requirements:

"The study was financially supported by private donations (Mrs. Abbigail Shaw and Mr. Davis

Schiller), the RUISSS McGill Centre of Excellence on Longevity and the Foundation of the

Jewish General Hospital (Montreal, Quebec, Canada). The funding sources have no involvement

in study design; in the collection, analysis and interpretation of data; in the writing of the

manuscript; and in the decision to submit the article for publication."

"No, the funders had no role in study design, data collection and analysis, decision to publish, or preparation of the manuscript."

5. Please amend the manuscript submission data (via Edit Submission) to include authors Kevin Galery, Christine Vilcocq,  Marc Afilalo, Olivier Beauchet.

Reviewers' comments:

Reviewer's Responses to Questions

**Comments to the Author**

1. Is the manuscript technically sound, and do the data support the conclusions?

Reviewer #1: Partly

2. Has the statistical analysis been performed appropriately and rigorously? 

Reviewer #1: I Don't Know

3. Have the authors made all data underlying the findings in their manuscript fully available?

Reviewer #1: No

4. Is the manuscript presented in an intelligible fashion and written in standard English?

Reviewer #1: No

5. Review Comments to the Author

Reviewer #1: This manuscript is meant to confirm the ability of the ER2 to predict hospitalization and length of stay in the ED and hospital in older adults admitted to the emergency department on a stretcher. This has been established in prior studies but not in Canada. It also focuses on maintenance of completion rates and time to completion as proxies for usability. I believe the manuscript has potential, even though it's primarily confirmatory, but it is currently limited by apparent inconsistencies in methods and presentation and somewhat awkward writing in places that makes it difficult to follow. My biggest concern is the apparent inconsistency between the methods and presentation of results, which causes confusion. I am also concerned about the apparent lack of reference group in regression models comparing outcomes among categories, which doesn't make sense.

Major concerns:

I’m confused by table 3 because every risk category has a parameter estimate. I would expect one of these to be the reference group against which others are compared. What is the reference group?

There are inconsistencies in the description of the statistical analysis and presentation of results. The statistical methods seem to indicate that both linear regression and Cox regression were used to study time to discharge from the ED and hospital. The description of the Cox model suggests there were two models, but table 3 suggests a Cox model was used to look at hospital admission as an outcome.

Figure 3 is described as time to discharge from the ED, while the statistical analysis section describes a Kaplan-Meier curve of time to discharge from hospital. Is this time to discharge from ED or hospital to a non-hospital location? I’m guessing so since this seems like the best strategy, but it’s not consistent with the presentation of the Cox model which seems like the related analysis or with how it’s described. How was death in the hospital or ED handled in this analysis? Were patients censored if they died?

Other issues:

I believe a more coherent story could be told if the rationale for certain measurements was stated up front, e.g. the completion rates and time to completion being proxies for usability. As I read I was confused as to the purpose of comparing these rates and times to completion by month. Even in the discussion it seems like the interpretation in relation to usability could be clearer. Also, the purpose of table 1 is never really clearly described. Late in the discussion section it's mentioned that the population may not be representative, without referring back to this table. Describing the purpose of various measures, creating this table, etc., would tell a clearer story.

I’m not sure I agree with calling a long LOS in the ED and hospital or hospital admission “short-term ED adverse events.” They are certainly outcomes of more severe conditions, but “adverse event” usually specifically refers to harm from medical care. I think something more generic like “worse outcomes” would be better.

In the results section of the abstract, I’m not sure why the betas for the high risk group are defined as >/=. Aren’t they just point estimates, such that this should be” =”? Also, presenting these as parameter estimates is not clear without specifying the medium risk group as the reference, but I remain confused about what these are being compared to since every group has a parameter estimate.

The clarity of writing could be improved in a number of places, e.g. on page 6 when describing high risk. The sentence starting with “An association of the two major items…” could simply state that one major item and at least one other major or minor item defines the high-risk level.

Why are the rates of ER2 completion and hospitalization described as ratios instead of just percents? Ratios seem like an unnecessarily complicated way of presenting the data. Figure 1 actually shows percents, so I’m not sure why the text and title suggest that the ratios are being presented.

Why did you choose to adjust for Canadian emergency department triage and acuity scale level in models? Were you looking for value added beyond this severity measure? That’s probably fine, but that could be discussed for context since both are getting at acute health status to some degree.

Was the analysis of LOS in the ED restricted to patients who were not hospitalized? It seems not. This seems like a good idea since very sick patients might move quickly to the hospital.

I believe “residences” is used to describe nursing homes or some kind of residential care. I think “residences” means wherever someone lives to most people, so different language for this category would be helpful.

6. PLOS authors have the option to publish the peer review history of their article (what does this mean?). If published, this will include your full peer review and any attached files.

Reviewer #1: No

---

## [Author Response · Author response to Decision Letter 0]

6 Nov 2020

Comment 1: “Thank you for stating the following financial disclosure: 

 [No, the funders had no role in study design, data collection and analysis, decision to publish, or preparation of the manuscript.]. At this time, please address the following queries:

We added the requested information in our revised manuscript. The study was self-funded by the Centre of Excellence on Longevity, through funds allocated to research by the Foundation of the Jewish General Hospital. The authors received no specific funding for this work. The funders had no role in study design, data collection and analysis, decision to publish, or preparation of the manuscript.

Comment 2: “We note that your Data Availability Statement states that “Effectively, there is an ethic restriction from the Jewish General Hospital (Montreal, Quebec, Canada) which approved the study to access to the data. So, Data and materials are available on request. Request should be sent to : Julia Chabot, MD ; Department of Medicine, Division of Geriatric Medicine, St. Mary's Hospital Center

3830 Lacombe Avenue, Montreal, Quebec H3T 1M5; E-mail: julia.chabot@mcgill.ca. All requests need a cover letter explaining the objective, justification and the referent ethic committee.”

Comment 3: “Please amend the title either on the online submission form or in your manuscript so that they are identical.”

The Editor is correct, we made the changes.

---

## [Decision Letter · Decision Letter 1]

3 Dec 2020

PONE-D-20-15983R1

Risk for short-term undesirable outcomes in older emergency department users: Results of the ER2 observational cohort study

PLOS ONE

Dear Dr. Launay,

Thank you for submitting your manuscript to PLOS ONE. After careful consideration, we feel that it has merit but does not fully meet PLOS ONE’s publication criteria as it currently stands. Therefore, we invite you to submit a revised version of the manuscript that addresses the points raised during the review process.

The Academic editor served as the second reviewer for the manuscript due to challenges with obtaining external reviewers.  On the bases of this review, there is agreement - there are number of editorial and reasoning details that have be addressed. The main reviewer outlines these very well, critical attention to these comments are warranted. In addition, it remains unclear what the research question is and how it is linked (needs more specificity) to the results and conclusion. If these are addressed the manuscript would be better considered in regards to publication.

We look forward to receiving your revised manuscript.

Kind regards,

Joseph Telfair, DrPH, MSW, MPH

Academic Editor

PLOS ONE

Reviewers' comments:

Reviewer's Responses to Questions

**Comments to the Author**

1. If the authors have adequately addressed your comments raised in a previous round of review and you feel that this manuscript is now acceptable for publication, you may indicate that here to bypass the “Comments to the Author” section, enter your conflict of interest statement in the “Confidential to Editor” section, and submit your "Accept" recommendation.

Reviewer #1: (No Response)

2. Is the manuscript technically sound, and do the data support the conclusions?

Reviewer #1: Partly

3. Has the statistical analysis been performed appropriately and rigorously? 

Reviewer #1: No

4. Have the authors made all data underlying the findings in their manuscript fully available?

Reviewer #1: No

5. Is the manuscript presented in an intelligible fashion and written in standard English?

Reviewer #1: Yes

6. Review Comments to the Author

Reviewer #1: This paper still suffers from many of the same issues as in the last submission. The models that lumped groups are better explained, but I still don’t understand the purpose of modeling in this way. Model 2 in table 2 should be adequate and makes a lot more sense than the models described as model 1. I don’t think the model 1 models add anything and believe that they detract by being confusing. There are other internal inconsistencies in the writing and presentation. Please make sure all p-values and comparisons are accurately described in the text, and are consistent with the tables and figures.

Page 5 and 9 describe the ratio of completed to not completed assessments as the percent, but this is not the percent. The denominator for a percent should be number of older adult ED users. The results also describe this inaccurately with respect to figure 1 if this is truly a percent.

Similarly, on page 9, hospital admission rate is described as proportion of patients admitted to the hospital divided by the proportion discharged to the ED. The denominator should be all patients. I also don’t know that discharged to the ED makes sense. I think it would be discharged to home or anywhere other than the hospital, but this is irrelevant since the denominator for a rate should be all patients.

I’m now more confused about what happened to people who died during their hospitalization. The description of the sample indicates that death during hospitalization was an exclusion criterion. The description of Kaplan-Meier curves now says it’s a censoring criterion. Which is correct? An outcome like hospital-free days over a 2-4 week period could have advantages in that it would capture death as a severe outcome (no hospital free days), but I don’t feel strongly about this. The discrepancy just needs to be addressed. I can understand that individuals who died would need to be excluded from certain analyses. Considering them to have a short length of stay due to rapid death would be a problem for the LOS analyses.

The p-values shown in figure 1 do not seem to be accurately reflected in the results. October vs. November is described as significant in the results, but the p-value provided is for October vs. December. This also appears significant. All told, figure 1 seems to suggest that completion rates were highest in October then declined slightly in subsequent months. The conclusion that completion rates increased is basically accurate when considering the lower rate of completion in the first month of implementation, but the numbers really seem to reflect an increase in month 2 followed by a decrease in month 3, with a non-significant trend toward another increase in month 4 that still lags behind month 2.

For the models in table 2, what is the purpose of including model 1, which is actually three different models, instead of just using model 2? Why would you merge low and high risk together as a reference group and compare the moderate risk group? That is intermediate vs. the other two groups, so the comparison makes little sense. I understand the sample size of the moderate risk group may be a concern that could lead with lumping with low or high risk, but model 2 uses a much more typical approach that is interpretable to readers, while model(s) 1 uses an unusual and confusing approach that adds little useful information in my opinion. I can’t think of any reason to include model(s) 1. If you’re interested in contrasts of high vs. medium risk, these could be examined as contrasts within model 2. The sample size for comparisons of the low risk vs. high risk group are more than adequate without lumping them with the medium risk group.

In table 2 footnotes, what does it mean that this was “Separated model for each Emergency Room Evaluation and Recommendation risk level”? Are you saying the model was stratified by this variable? If so this should be described in the methods. It does raise some questions about why adjustment was inadequate and whether there was effect modification by risk level, but overall I think a stratified model is reasonable.

Minor comments:

The abstract is confusing because of the way the results are presented. The lengths of stay in hours look like actual lengths of stay instead of the difference in length of stay. Please rewrite this for clarity.

I don’t understand the meaning of “hospital health areas.” This needs to be described somewhere.

There are occasional issues with grammar and usage in the paper. Careful review of the paper by someone with English as a primary language would be helpful, though many of these issues could be addressed at the text editing stage.

7. PLOS authors have the option to publish the peer review history of their article (what does this mean?). If published, this will include your full peer review and any attached files.

Reviewer #1: No

---

## [Author Response · Author response to Decision Letter 1]

29 Jan 2021

Response has been attached in the files

---

## [Decision Letter · Decision Letter 2]

23 Feb 2021

PONE-D-20-15983R2

Risk for short-term undesirable outcomes in older emergency department users: Results of the ER2 observational cohort study

PLOS ONE

Dear Dr. Launay,

Thank you for submitting your manuscript to PLOS ONE. After careful consideration, we feel that it has merit but does not fully meet PLOS ONE’s publication criteria as it currently stands. Therefore, we invite you to submit a revised version of the manuscript that addresses the points raised during the review process.

The editor remains  as the second reviewer for this manuscript. I'm not sure if the statement that one can email the author with proper approvals to get the data is adequate to meet the data sharing policy requirements.  Not including the original data with ages can be understood since some of them are identifiable (90 or older).Suggest the key data elements used in the manuscript could be supplied deidentified with categorical ages if public sharing is crucial, e.g. 65-74, 75-84, 85+.

The English usage still needs a little work but it's mostly okay. This manuscript could be improved with text editing.

The authors are strongly encouraged to address concerns of both reviewers to be further considered for publication.

We look forward to receiving your revised manuscript.

Kind regards,

Joseph Telfair, DrPH, MSW, MPH

Academic Editor

PLOS ONE

Journal Requirements:

Reviewers' comments:

Reviewer's Responses to Questions

**Comments to the Author**

1. If the authors have adequately addressed your comments raised in a previous round of review and you feel that this manuscript is now acceptable for publication, you may indicate that here to bypass the “Comments to the Author” section, enter your conflict of interest statement in the “Confidential to Editor” section, and submit your "Accept" recommendation.

Reviewer #1: (No Response)

2. Is the manuscript technically sound, and do the data support the conclusions?

Reviewer #1: Yes

3. Has the statistical analysis been performed appropriately and rigorously? 

Reviewer #1: Yes

4. Have the authors made all data underlying the findings in their manuscript fully available?

Reviewer #1: No

5. Is the manuscript presented in an intelligible fashion and written in standard English?

Reviewer #1: Yes

6. Review Comments to the Author

Reviewer #1: The manuscript presentation is improved but there are still a few issues that need attention.

Abstract

The results in the abstract appear to reflect the old models that have been removed from the paper, as they use both other groups as the reference. Please update these to reflect the results in the new table 2 (medium vs. low and high vs. low).

Discussion:

I don’t see strong support for the following sentence.

“It suggests that older ED users who will stay a shorter period of time in ED or in hospital have a phenotype of middle-age patients with few and less severe acute diseases and minimal to no disability.”

I get the idea that the lower risk patients were possibly less likely to be old-old but calling them middle-age makes one think of a middle-aged person. I don’t think any patients in this study would be classified as middle-aged per the usual definition. The age difference also isn’t that large. A mean age of 83 in the low-risk group doesn’t seem like a middle age vs. others. I don’t see categorical age in table 1, so it’s hard to phenotype patients in terms of their likelihood to be a particular age. They’re generally slightly younger, but that’s about all that can be concluded.

Similarly, I don’t think data are presented to support the idea that they have minimal to no disability. Only use of walking aids and use of home support are included in the tool, and the latter is a minor criterion that could leave someone in the low-risk group. There are many kinds of disability beyond need to use walking aids.

Also, what data supports that they have less severe acute diseases? This is perhaps reasonable to assume because of the lower likelihood of hospitalization, but it doesn’t seem to have much support from table 1. There are only limited differences in the triage scale and they sometimes suggest low scoring patients were more acutely ill in some respects, even if they were less frail. The reasons for the ED visit don’t vary that much. It could be that frailty matters as much or more than the conditions themselves, at least as they’re classified in the data presented.

Minor comments:

Page 18. It's a little awkward to write in this sentence that the low-risk group was the reference group. That's noted in the methods. “In addition, regarding prolonged LOS, using the low-risk group as reference showed that the high-risk group was associated with a longer LOS in ED…”

I think this would read better if it just said LOS in the ED and hospital was longer in the high-risk compared to the low-risk group.

A few grammatical issues remain but I think they can likely be handled in text editing.

7. PLOS authors have the option to publish the peer review history of their article (what does this mean?). If published, this will include your full peer review and any attached files.

Reviewer #1: No

---

## [Editor Report · Decision Letter 3]

29 Mar 2021

Risk for short-term undesirable outcomes in older emergency department users: Results of the ER2 observational cohort study

PONE-D-20-15983R3

Dear Dr. Launay,

We’re pleased to inform you that your manuscript has been judged scientifically suitable for publication and will be formally accepted for publication once it meets all outstanding technical requirements.

Kind regards,

Joseph Telfair, DrPH, MSW, MPH

Academic Editor

PLOS ONE
---

## [Editor Report · Acceptance letter]

2 Aug 2021

PONE-D-20-15983R3 

Risk for short-term undesirable outcomes in older emergency department users: Results of the ER^2^ observational cohort study 

Dear Dr. Launay:

I'm pleased to inform you that your manuscript has been deemed suitable for publication in PLOS ONE. Congratulations! Your manuscript is now with our production department. 

Kind regards, 

on behalf of

Dr. Joseph Telfair 

Academic Editor

PLOS ONE